# Ragging as an expression of power in a deeply divided society; a qualitative study on students perceptions on the phenomenon of ragging at a Sri Lankan university

Ayanthi Wickramasinghe[¤][☯]*, Pia Axemo[¤][☯], Birgitta Essén[¤], Jill Trenholm[¤]

International Maternal and Child health unit, Department of Women and Children's health, Uppsala University, Uppsala, Sweden

☯ These authors contributed equally to this work.
¤ Current address: Department of Women and Children's Health, MTC-huset, Uppsala, Sweden
* ayanthi.wickramasinghe@kbh.uu.se

**Data Availability Statement:** All relevant data are within the paper and its Supporting Information files.

## Abstract

Initiation rituals such as hazing, bullying, and ragging, as it is referred to in Sri Lanka, is a global phenomenon and has become a serious public health problem. Students are bullied and harassed by senior students causing them to suffer severe adverse consequences including depression, increased university dropouts and suicide. Although this has led to a significant burden on the country, research on ragging is scarce. The aim of this study was to explore the perceptions of students concerning the phenomenon of ragging and to understand how ragging affects student life and culture at the University of Jaffna, Sri Lanka. This paper is based on 17 focus group discussions with male and female students of Sinhalese, Tamil, and Muslim ethnicity. Thematic analysis was employed to navigate through the theoretical lenses of structural violence, intersectionality, and social dominance. The findings revealed how students perceived ragging differently; as an expression of power to initiate order and as a way to express dissatisfaction towards social inequalities occurring within the larger society or to facilitate bonds between university students. Students trivialized violence related to ragging and accepted it as a part of the university subculture despite being aware of the dire consequences. There was a described cyclical nature to ragging whereby victims become perpetrators. The student's perspective appeared to be a missed opportunity in finding feasible solutions to a societal problem that must take all parties involved, into consideration.

## Introduction

Many university students look forward to gaining entry to a university, and celebrating this milestone in their transition to adulthood. However, this elation and excitement is often replaced by fear and anxiety when students have to undergo harmful initiation practices and rituals. Although the terminology differs from country to country; "hazing" or "bullying" in

**Funding:** This research was funded by the Faculty of Medicine at Uppsala University.The funders had no role in study design, data collection and analysis, decision to publish, or preparation of the manuscript.

**Competing interests:** The authors have declared that no competing interests exist.

USA, "bizutage" in France, "praxe" in Portugal and "Mopokaste" in Finland, there are commonalities to these practices found in higher education institutes and universities around the world [1]. In Sri Lanka and most South Asian countries, this so-called initiation ritual is known as "ragging". It is carried out by senior students and was originally created to forge comradery and bonds of friendship. However, ragging has evolved in South Asian countries to have little remnants of its original form [2–4]. In Sri Lanka, ragging is defined as "any deliberate act by an individual student or group of students, which causes physical or psychological stress or trauma and results in humiliating, harassing and intimidating the other person" [5]. It has been expressed as an intentional and systematic violation of human rights and freedom of thought and movement of the junior students [6].

Depending on the country and cultural contexts, initiation rites differ as does the goal of the practice. In the western world, hazing practices, especially within fraternities and sororities, mostly consist of sexual abuse, drinking games including forced binge drinking in order to form social bonds [7–9], whereas ragging practices in Sri Lanka are built upon breaking cultural taboos often seen as an opportunity to equalize students from different societal backgrounds [10, 11]. These practices, seen differently in different countries can be viewed as highly contextual, often mirroring the society at large and exhibiting different power dynamics related to race, gender, socioeconomic status and other facets of student's identity [12]. Ragging in Sri Lanka has been considered to be distinct as it has been shown to be driven by an outcry of discontent towards authoritative figures and societal hierarchies [13, 14].

To put Sri Lanka in context, it is a multicultural, multilingual country consisting of an ethno-religious blend of Sinhalese (75%), Sri Lankan Tamils (11%), Moors (Muslims) (9%), and other groups (5%) [15]. The Northern and Eastern Provinces are predominated by Tamil Hindus, and the rest of the country is predominated by Sinhalese Buddhists. The official languages are Sinhala, Tamil, and English, a remnant of colonialism. This post-war nation is still struggling from its history of colonialism [16] and the 27-year long civil war which ended in 2009. Sri Lanka is a country rich in traditions, still believed to uphold patriarchal values and a hierarchical social structure [17].

Following the colonial rule, the Sri Lankan educational system changed from an elitist model, where only the wealthy partook, to a mass model where all citizens were welcomed in higher education. Along with this, the change in the medium of instruction from English to Sinhalese and Tamil, led to changes in the composition of the student population [14]. The most prominent feature of mass university education was the change in socioeconomic structure of the student population due to the district quotas enabling a higher intake of rural students who are often from poorer backgrounds [18]. The previous majority of English-speaking urban upper and middle class was replaced by Sinhalese and Tamil speaking lower classes from lesser privileged backgrounds [18].

The lack of adequate English skills in this new cohort, has hindered these students from eventually securing so-called desirable jobs and entering the global job market. The resultant high unemployment rate has led Sri Lankan youth to feel frustrated and perceive the country to be unjust, unequal and marginalizing [19]. Therefore, these culturally embedded underlying inequalities have become a breeding ground for dissatisfaction and have contributed to the changing practice of ragging. Ragging practices are carried out by the "seniors", who are students from the second year and above, forcing the new entrants to university to conform to their rules. Ragging often entails the newcomers being subjected to psychological, verbal and physical abuse such as beating, hitting with objects, performing dangerous tasks, and sexual abuse such as stripping, performing sex acts and rape [1, 10]. This has led to a range of health consequences like anxiety, depression, insomnia, injuries, and even death and suicide [20]. Ragging has become a significant public health problem which not only has led to ill health,

but has resulted in a loss of students from universities, with a subsequent loss of human resources and economic prosperity in the country [13].

Ragging has been a criminal offense in Sri Lanka since 1998 and carries a severe punishment [21], yet this has not deterred students from carrying out this ritual covertly. This practice is embedded as part of the university 'subculture'[17]. A recent report conducted among students in eight Sri Lankan state Universities found the prevalence of verbal ragging to be over 51%, psychological ragging 34%, physical ragging 24% and sexual ragging 17% [22]. Initiatives made by the University Grants Commission (UGC), the administrative body of universities, by issuing guidelines [5] and creating several methods to lodge complaints against ragging but these initiatives have not been successful in curbing this practice [10].

There is a scarcity of research on ragging in Sri Lanka, particularly around student's perceptions. According to reports by the Ministry of Education, approximately 2000 students dropout annually, and several students have committed suicide as a result of ragging [23]. Similarly, a study conducted in Bangladesh demonstrated, traumatic incidents such as ragging increased suicidal ideation among university students [24].

Educating youth in a safe space is essential, particularly for its subsequent contributions towards the country's future. It is increasingly imperative to address this serious public health problem that profoundly affects all students, not only victims but also perpetrators and bystanders. Ragging has a potential deleterious impact upon society's younger generations and their university years intended for building intellectual capacity. The aim of this study was to explore students' perceptions concerning the phenomenon of ragging, and to understand how ragging affects student life and culture at the University of Jaffna.

## Integrated theoretical lenses

Universities are microcosms of the larger society [14]. Contexts matter and ragging communicates within this complexity, deeply affecting students' lives and behaviour. The following integrated theories, explained below, helped sensitize us during the data analysis.

Galtung's theory [25] of structural violence was the theoretical departure point at the macro level. This theory holds that direct violence, like ragging, is the visible manifestation of underlying invisible violence that goes unquestioned in everyday praxis. According to Galtung, structural/instutionalized violence reveals how societies naturally purvey their cultural beliefs veiling the reality of destruction by making violence seem acceptable.

The essence of intersectionality [26, 27] is inextricably linked to structural violence. By examining the diversity of the students who make up this microcosmic society of the university, one notes how personal identities and their many other diverse attributes influence beliefs, actions, and experiences. How these cultural features/attributes intersect, renders the individual more than a sum of their parts. Intersectionality reveals the individual's many unique intersecting factors, such as student's age, gender, ethnicity, social class, caste, language, cultural history, and geographical origins, giving a more nuanced view of their marginalized states.

These individual factors, not only build upon their identity, but also allocates them into various social groups that mirror the hierarchies of the larger society. This leads to group formations that attempt to wield power/status, enabling them to control other groups. This group dynamic is in alignment with Social Dominance theory, which states that dominant individuals have a tendency to organize themselves into ingroups and outgroups to form social hierarchies, with the ingroup dictating or controlling the outgroup [28].

Structural violence, Intersectionality and Social Dominance theories are integrated, and serve to reflect the macro, meso and micro levels of society. Employing this matrix of theories, the complexity and subtleties of ragging are further illuminated (Fig 1).

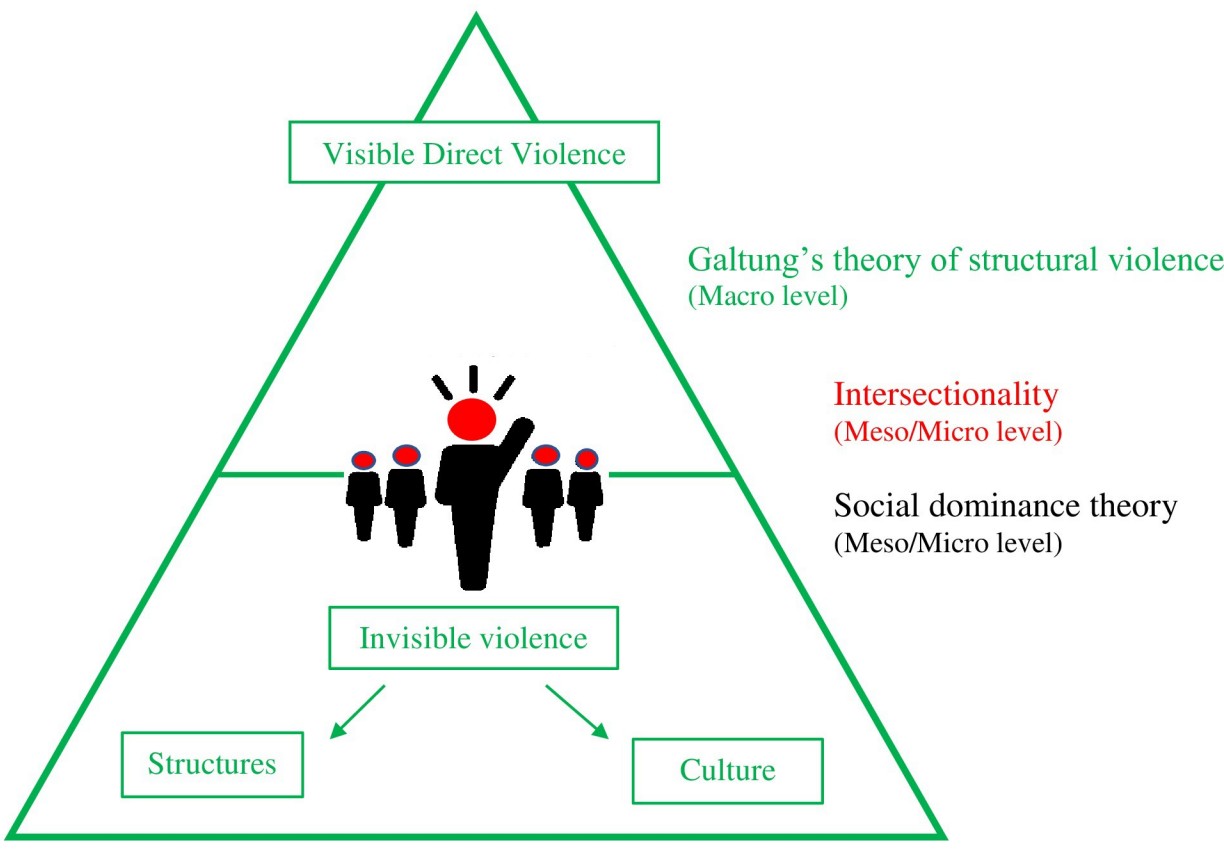

**Fig 1. Integrated theoretical lenses.**

## Materials and methods

### Study design

A qualitative phenomenological design was used for this exploratory study. Focus group discussions (FGDs) were chosen as it is the method of choice when exploring societal/group norms, revealing their lived experiences and how attitudes and behaviors are formed within groups [29].

### Study setting

The University of Jaffna, is situated in the Northern province of Sri Lanka. Jaffna was a major war zone during the ethnic conflict between the Sri Lankan government and the Liberation Tigers of Tamil Eelam that ended in 2009. A majority of the population of the Northern province are Tamils [15]. Since 2012, similar numbers of students of all ethnicities began their studies at the University of Jaffna. Most other universities across Sri Lanka have a Sinhalese majority. The university of Jaffna was chosen for this study as the student composition was more diverse.

### Study participants

The principal investigator (PI)/first author, approached heads of departments from Medicine, Arts, Management, Science, and Technology faculties, and asked them to inform students enrolled in these faculties about the study and how to contact her if they were interested in

participating in the FGDs. From the students who responded, a convenient sample of 50 male, and 58 female students, from the 2nd and 3rd year, between 21 to 25 years were selected. The chosen participants could include victims, perpetrators and by-standers. First year students were excluded as they could suffer re-traumatization as they could have most likely experienced ragging recently. The participants were informed about the study by the PI and explained that it was a part of a larger doctoral project involving ragging.

The PI is a Sri Lankan born medical doctor currently undergoing her doctoral studies. She has grown up in the context yet received her own education overseas. Having not attended Sri Lankan university yet being part of the culture, she has a distinct insider/outsider vantage point. She spent over a month in Jaffna prior to data collection to get acquainted with the university setup and familiarize herself with the surroundings.

## Data collection

A total of 17 FGDs were conducted with individual groups of Sinhalese, Tamil, and Muslim students, organized by ethnicity, language, and gender as it was deemed more appropriate and for ease of communication amongst like students. Similar numbers of FGDs were conducted with each ethnic group/gender. Discussions were carried out in three languages. The English FDGs were moderated by PA, a Swedish medical doctor with extensive experience in cross cultural collaborations and longstanding work history in Sri Lanka. The Sinhalese FDG's were moderated by KW a Sri Lankan researcher with expertise in qualitative methods. Some of the English and Sinhalese FDG's were moderated by the PI who is Sinhalese speaking. A Sri Lankan Tamil speaking research assistant moderated all the Tamil FDG's. Notes were taken by two observers in the Tamil focus groups, and for the other groups, one observer was present. A thematic guide was used, with questions related to student's perceptions of ragging, perpetrators, victims, campus environment and student recommendations. Observers notes enriched the data collection and were used in debriefings. Groups consisted of four to eight participants, with each discussion lasting 45–60 minutes. A quiet location that ensured privacy and located on the university premises was used. All FGDs were carried out in March 2019, except for the Faculty of Technology, which was conducted in November 2019 due to the closure of the faculty concerning a severe ragging incident [30].

## Data analysis

The participating researchers discussed the FDG's immediately after they were conducted, prior to conducting subsequent FGDs in order to refine probing questions and to incorporate emerging information. FGDs were transcribed and translated/back-translated into English by the PI and a Tamil speaking research assistant. This process contributed to enhancing familiarity with the data as transcripts were read repeatedly during the transcription and translation processes. Transcripts were analyzed by the PI and two other researchers with both insider and outsider positions, using thematic analysis [31]. The research team initially coded the themes independently and then met and spent several days coming to a consensus, mapping, defining, and redefining the themes. Notes concerning background information, comments, and innuendos were used to better understand and substantiate the material. Through the chosen integrated theoretical lenses; Structural violence [25], Intersectionality [27] and Social Dominance [28]. The transcripts were iteratively read and inductively coded. Subsequently, similar emergent codes were grouped together in a mapping exercise. Quotes were used to enhance credibility. Themes and sub themes were developed from the multiple data interpretation discussions. During this process, one overall theme and four main subthemes were chosen by consensus of the research team.

### Ethical considerations

Ethical approval was granted by the ethical review committee of the University of Jaffna, Sri Lanka (J/ERC/18/96/NDR/0200). The PI provided information about the aim and procedures of the study to the participants and obtained a written informed consent before starting the interviews. Confidentiality and anonymity were ensured by assigning each participant a code according to ethnicity and gender, Sinhalese (S), Tamil (T), Muslim (M) and male (M), and female (F), which was used to identify the transcripts.

## Results

Ragging as an expression of power was established as the overall theme (Fig 2).

There were five subthemes as follows; veil of secrecy and silence, ragging lies on a spectrum, cycle of ragging establishes a hierarchy, a society with deep divisions and student recommendations; an unexplored potential resource. Four of the subthemes focused on inter-student relationships and their dynamics, while the other, portrayed the complex interaction with teachers, university administration and society.

### Subtheme 1: Veil of secrecy and silence

The existence of ragging within the university was expressed as a well-kept secret among the students. "No, it never happened to us", was how most FGD's began with students claiming that

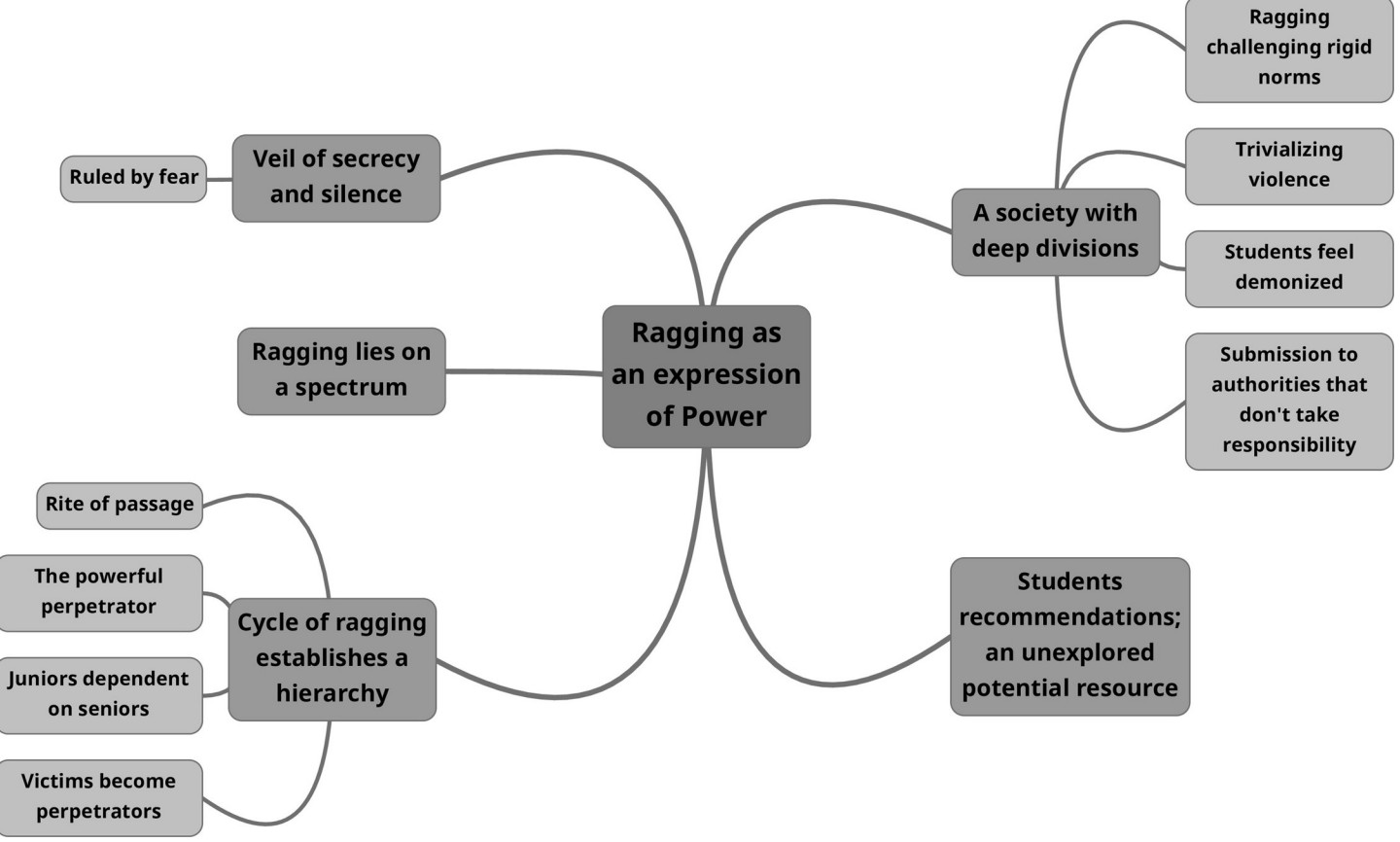

**Fig 2. Main theme and subthemes derived from the analysis of student focus group discussions.**

ragging did not occur in the University of Jaffna and that they had never heard, or experienced it. They ultimately contradicted themselves when they then went on to explain how it was.

Students claimed that the seniors gave the newcomers humiliating names which dehumanized them, and the seniors themselves were identified by pseudonyms that demonstrated power, keeping them "nameless", further maintaining the secrecy of this ritual.

"They (seniors) will put names. The names will have very bad meanings. For the next four years they will only use that name." (T, F)

**Ruled by fear.**   It was evident that generating fear and intimidating newcomers was the way the seniors maintained secrecy. One student put it this way:

"First years are afraid. They won't talk about ragging to anyone. Even if we (batch mates) ask who carried out the ragging, they won't tell." (T, M)

Other students thought the raggers threats took away the juniors' independent decision-making capacity, suppressing their ability to differentiate right from wrong expressed here;

"Introducing them (juniors) to a new place and showing them how to behave by reducing their capacity of self-thinking and decision making from their own experience, creates a hive mentality within the students." (T, M)

## Subtheme 2: Ragging lies on a spectrum

For many students ragging was part of the university "subculture" which the newcomers faced in their first year, and sometimes longer. Ragging practices were seen as both positive and negative, which can be as mild as singing or a dress code to extortion and violence. The junior women had to wear dresses made from a certain fabric generally worn only at home, referred to as "Cheeta dresses" in Sri Lanka and braid their hair in two braids, seen as juvenile for their age group. The men had to shave their heads, wear white long-sleeved shirts, no belts, and no underwear. Both men and women had to wear a certain type of bathroom slippers and were not allowed to wear shoes. This was meant to humiliate and infantilize them. This student expressed the stigma related to being dressed in bathroom slippers;

"They (seniors) treat them (newcomers) like small children. . .when they go in the bus people will laugh at them . . .are you working in the bathroom, like bathroom cleaners?" (M, F)

The victims of ragging were said to be the junior students, and the ones that were picked on the most, were men that were "handsome", came from so-called "good schools" and did not "respect the seniors". The seniors who engaged in or instigated ragging, were in most cases students who had undergone ragging as juniors, had "under par academic performance", "violent tendencies" and/or "inferiority complexes". In their words, a ragger was;

"A person who came from a low-level society, studied in a low-level school, and had nothing special in his life and wanted to enjoy everything here." (M, M)

The students spoke about both the negative and positive consequences of ragging; "not all ragging becomes violent". While other students said that "Ragging and interaction are being

used as synonyms" by the seniors to justify ragging. This demonstrates how ragging was carried out using greater and lesser degrees of violence/coercion, from things such as asking the newcomers to sing a song to assaulting the newcomers.

The persistent harassment of new students was a recurrent topic in the FDGs as is described here;

"Before and after every lecture the seniors will come to the lecture hall and rag us. Every hour at least 10 minutes they will rag us." (T, F)

Even though ragging took a predominantly psychological or physical form, there were instances students were subjected to sexual forms. Students were reluctant to expose the occurrence of sexual harassment but the following was revealed;

"Some kind of touching (sexual) also goes on. . . Sinhalese girls are sexually harassed." (M, M)

Another student said;

"They (newcomers) have to imitate dogs having sex. Girls are asked to draw boy's body parts, and the boys to draw girl's body parts." (M, F)

Several students said ragging was necessary. They believed it was a positive way to develop bonds and increase comradery between the seniors and juniors.

"When we come to this campus, those are the friendships that remain with us. To have a friendship we need to form a connection, because of ragging a connection was formed." (T, M)

Some students believed that ragging should continue in a non-violent manner because it helped develop and nurture new skills such as communication;

". . .with a thing like this (ragging), their (newcomers) personality improves, this is my experience, before I wasn't a person who would speak out like this but little by little it improved" (S, M)

Another argument made by the students concerning positive ragging, was that seniors perceived they helped newcomers by guiding them towards the "correct path" using fear, demonstrated in the following quote:

"If we tell them (junior females), maybe in fear they will wear (culturally) appropriate clothes from the first year. If not they will face problems when they go out (society). So, we can't completely stop ragging." (T, F)

The participants mentioned that seniors often kept juniors awake throughout the night and woke them up early in morning to rag them, thus causing sleep deprivation and exhaustion. Therefore, newcomers lacked time for their studies and were frequently drowsy during lectures.

"We (newcomers) are told to come at 9.00 pm and they (seniors) let us go around 12 midnight. Even if we say that we are feeling very sleepy, they don't listen. If you're sleepy you have to sleep on the floor, we have to wait till they (seniors) let us go." (M, F)

### Subtheme 3: Cycle of ragging establishes a hierarchy

This sub-theme revealed that ragging was a cyclical system organized to create a hierarchy within the university student body by the demonstration of 'power over'.

**Rite of passage.** The FGD participants saw ragging as a rite of passage that all students entering university must undergo to be accepted by their seniors and belong to their peer group. The "ragging period" ends, with the so-called "ponding ceremony" where newcomers are thrown dirty stagnant water on. Following this, the seniors give the juniors a "welcome party" where the seniors and juniors unite as "Batch fit" (an expression that indicates belonging to the group).

> "Ragging is like an acceptance to campus. When we give the first-year students the welcome party then there is some unity." (S, M)

Students stated that it was a ritual with a few unwritten rules such as, ragging occurs only among your own faculty and only seniors can rag. Men were said to be ragged by men, and women ragged by women, and most importantly, ragging occurred strictly within ethnicities.

> "Other faculty seniors should not hit our juniors. Only we will hit our juniors." (T, M)

**The powerful perpetrator.** Several students referred to ragging as a way for seniors to demonstrate power and seniority, exhibiting their power over the juniors in the following manner:

> "You are a newcomer, I am going to show you my superiority by ragging, physically, mentally and psychologically." (T, M)

Another student spoke of how ragging was believed to prepare them for the greater world:

> "He (senior student) said ragging is to improve leadership qualities. If you don't obey your senior, how will you obey your boss in future?" (M, M)

Students who were disrespectful towards the seniors, got ragged more. Newcomers who made complaints, were often isolated from their batch, branded as "anti-raggers" and stigmatized. They were said to be excluded from university functions, parties organized by the students, not given leadership roles, and frozen out by the rest of their peers.

> "If someone won't obey them (seniors), they will separate that person from the batch and won't involve that person in common events." (M, F)

Students spoke about the financial burden caused by ragging. The seniors were said to extort money from newcomers, asking them to buy food for them in the cafeteria or to top up their mobile phone accounts. Some students, gave up their food for the day to fulfill the demand of feeding the seniors as indicated here;

> "Every student comes from a different economic status. I have seen some students (juniors) buy them (seniors) food and then they don't eat for the whole day or they will eat only one time per day." (T, F)

**Juniors dependent on the seniors.**   Juniors were said to be forced to depend upon seniors for academic support, as the lectures were at times inadequate or the students did not understand the lecture material due to language problems.

"They (seniors) do it (extra classes) willingly, as a help to the junior students." (S, F)

**Victims become perpetrators.**   Participants claimed that new students got revenge by ragging newcomers the following year. Students expressed that they did not see the harm in ragging their juniors as they themselves underwent it.

"If a boy got ragged, in future he will think about how he could develop that ragging and give it back. They (the boys) will talk in which way they could make it worse." (T, F)

## Subtheme 4: A society with deep divisions

Students spoke about how rigid norms and hierarchies in Sri Lankan society influenced ragging. Participants spoke about their tenuous relationships with authority figures which seemed to be imbued with fear. There was talk of distrust on both sides, authority figures and students. Ragging rules exist but action was expressed as rarely taken.

**Ragging challenging rigid norms.**   Participants alluded to Sri Lankan society's deep divisions in age hierarchies, gender, ethnicity, language, socioeconomic class, and caste which had an impact on ragging.

It was evident that the students' behavior was guided by traditional gender norms, displayed by the soft-spoken demure manner in which women spoke as opposed to the more loud, aggressive manner the males expressed their opinions. During the FGD's, the women often giggled and whispered when certain topics were discussed, whereas the men got agitated and angry when the female moderators probed their views.

They expressed how ragging occurred as per these gender norms. Men used more physical ragging to show off power to impress women as they did not often get opportunities to interact freely.

"Boys will hit handsome (male) students. Because he (ragger) can impress juniors (females)." (T, F)

According to the students, ragging practices were gendered, in that men underwent more physical ragging than women;

"They (seniors) will make us (females) sing and they will beat the male students." (T, F)

Senior women considered it their responsibility to guide and make sure the junior women upheld their respectability. Controlling the newcomer's behavior was a way to safeguard their decorum. A female student commented:

"I saw a girl sitting on a final year student's bench after 6pm and laughing with senior boys. The boys will be drunk. The girls don't know what will happen to them. When we saw that we got angry. They should think about their protection. They should think, how will I go back home safely." (T, F)

It was expressed how male students were often drunk in the evenings. Newcomers were said to be forced to consume alcohol, as it was seen as being unmanly if they did not. The cultural norm states that women are not supposed to drink alcohol, whereas men were defined by drinking. This was elaborated by the following;

"They will force the male newcomers to drink if they don't, they will scold them. . . you are not a boy, you are a girl." (M, F)

The students saw ragging as a way to equalize divisions such as socioeconomic status and caste, while increasing divisions in other cases.

". . .they (juniors) are brought together. Ragging reduces the disparity in the different levels of society and brings everyone to the same level. . .We bring them all down to the same level." (S, M)

The senior students felt it as a part of their duty to equalize everyone and "fix" the so-called mentality of the more privileged to become more equal with the more marginalized groups.

"We call them privileged. . .We take them and bring them all to the same level, and we fix their mentality." (S, M)

The participants felt very strongly about it being unfair that certain students who came from a stronger financial background could afford better things than others. These discussions became quite heated;

". . .If I go to the canteen with 60 rupees and he has 100 rupees. . .I can eat a vegetable rice, he can have a fried rice and drink a coke cola. We are in the same batch, we sit for the same exam, we study in the same campus and we study together, then why is there a difference?" (S, M)

The lack of knowledge of the English language led to a lot of difficulties among the students in communicating with students and lecturers of other ethnicities, understanding lectures and basic interaction with people living in the area. Students expressed that the English language was seen as a class marker, creating a social divide, as evidenced by;

". . .Some students try to show a fake poshness (by speaking English) . . ." (S, F)

Not allowing juniors to speak English during the ragging period was another method of suppressing students;

"In the first semester we have to talk using only Sinhalese words, we are not allowed to speak a single word in English." (S, M)

**Trivializing violence.** Many students believed that ragging be a part of university life/culture and they did not see the harm. They convinced themselves that it was harmless despite contrary information. This notion is reflected by this statement:

". . .but unlike other campuses there isn't ragging here. But by ragging no one is mentally or physically hurt here. If it's done there is no harm caused to anyone. . ." (S, F)

Some students felt that violence was a part of the Sri Lankan society and that people had to be obedient to the hierarchical systems, therefore they did not see ragging as something to be concerned about.

"In every situation people should have obedience. We should obey someone, maybe in an office. You should obey your boss. When a person can't obey, he is subjected to violence." (T, M)

**Students feel demonized.**   Due to controversial ragging incidents, students lamented that all students were seen as perpetrators and portrayed in a negative light by the media, and therefore mistreated by the university authorities by false accusations.

"The ones who were not involved got an inquiry . . . they (university administration) wanted to put a noose around their neck. . . that person was not even involved in anything (ragging). . ." (S, F)

Another student expressed how complaints of ragging were demoralizing;

"When you're trying to educate us, why are the people above us trying to put us down by talking about ragging, ragging, ragging?" (S, M)

Some participants claimed that they were worried about being a part of the FGDs because they thought it was an inquiry where they would be falsely penalized for ragging;

"To be honest, none of us wanted to come here (for the FGD) today. We thought it's like an inquiry but when the science lecturer explained. . . we thought, let's go and tell our problems, we can't suffer like this every day." (S, F)

**Submitting to authorities that don't take responsibility.**   Although there were several ways to make complaints, students conveyed their frustration, that authorities often did not take any action. The students expressed disappointment that lecturers, counselors, and others responsible, did not want to intervene or get involved with ragging leaving a vacuum, where raggers rule.

"The lecturers stay aside and let ragging occur. The lecturers won't get involved, then the seniors behave the way they want and rag the juniors." (S, F)

### Subtheme 5: Students' recommendations; an unexplored potential resource

The students had several recommendations on how to end ragging. Some students believed that ragging was unnecessary but organized interactions between senior and junior students were needed. They believed that the use of these terms interchangeably was the main problem and evident in the statement below;

"Ragging and interactions are being used as synonyms. Those are two different words. Ragging is hurting someone for ones' entertainment. Interaction is creating a place to get connected with students from different areas and societies. It becomes a problem when these terms are used interchangeably. Students get confused with these two words but they are two distinct things. Interaction is needed but not ragging." (T,M)

Participants had ideas on how to help students respect one another through mentorship and how to make ragging a more harmless way to interact with each other. One of the recommendations was;

"Until we eradicate the mentality of the seniors to suppress their juniors, we can't eradicate ragging. To do that mentorship is important. Mentorship by lecturers, to tell me what university is all about, what are my rights, how can I reach help and security. This message should reach the new students before the seniors capture them. . ." (T, M)

It was expressed that during, wartime, the students were a more intricate part of the greater society and thus were more community oriented, versus the current more individualistic society;

". . .students (before entering university) were living inside a bubble created within school, home, and tuition classes. I think they need to think about society, they should make efforts to get connected with society." (M, M)

Anti-raggers were cited as a potential counter force, provided they got more support from students and the administration;

"There are anti-raggers in every batch. We can form a group through the university administration with the anti-raggers to identify raggers and to give them a punishment or suspension." (T, M)

## Discussion

This study expands the knowledge on students' perceptions of ragging and how ragging affects student life and culture in Sri Lanka. It was striking that the students themselves were ambivalent in their views of ragging. However, ragging used as an expression of power permeated the findings as an overall theme. This overall theme consists of five subthemes as follows; veil of secrecy and silence, ragging lies on a spectrum, cycle of ragging establishes a hierarchy, a society with deep divisions and student recommendations; an unexplored potential resource. These finding can contribute to a deeper understanding on how this negative ragging practice can be curbed and/or promote change in preserving the more positive experiences of bonding.

### The wider context of ragging: Sri Lanka and it's university culture

This study underscores Sri Lanka's historically embedded rigid social norms and hierarchical systems lending itself to youth's discontent which manifests in ragging as direct violence [25]. Beneath the facade of this purported equitable society, students revealed hierarchies of socioeconomic classes, caste, ethnicity and gender which has led to a clash of attitudes, differences in privileges, and perceived unjust divisions of power [32]. The diverse mix of students at Jaffna University seem to experience being re-grouped according to a hierarchical system whereby the seniors' ruled over the newcomers. This aligns with social dominance theory [28] and the stated administrative apathy creates a vacuum which contributes to the institutionalizing of structural violence. Samaranayake et al. [18] showed how students who feel alienated, ignored and unheard, by adult power structures both in the university and in society often turn towards violence; ragging can be seen as an expression of this violence. The students in this study shared the sentiments that they had to comply with the hierarchical restructuring and felt more let down than supported by the administration.

The English-speaking urban middle class continues to rise to the top in the national and private sector while sidelining the rural, monolingual, lower classes that are dependent on very limited government and public sector jobs. [11, 19]. Despite common dissatisfaction, disgruntled youth of different ethnicities remain isolated from each other due to segregation by language divisions and regional barriers [16]. Gamage et al. [16] remarks that ragging has served as a method of ensuring those from privileged backgrounds are made aware that they are not superior to those from less privileged settings, thereby disrupting the existent societal norm. Similarly, this study's findings demonstrate how ragging is a tool to equalize societal hierarchies and associated disadvantages giving voice to the marginalized, by leveling the playing field. This was evident in how the participants spoke about "equalizing" students. The social injustices faced by the youth can be considered as invisible structural violence as per Galtung [25]. It has been shown that students' involvement in confrontational politics, could be an attempt to empower themselves, while the Marxist and leftist political parties are said to be the driving force behind student unions eager to recruit dissatisfied youth [18].

Intersectionality reveals the gendered dimension found in our study. Differences in the methods of ragging between men and women were further strengthened through patriarchal-driven norms repeatedly endorsed by society. It was evident that university students still upheld the traditional notion of '*Læjja-baya*' (Shame-fear), expected of women. Every child in Sri Lanka has been taught to conform to the concept of '*Læjja-baya*' or in other words, 'shame and fear of ridicule' which is exploited in ragging [17]. Young women are expected to behave with sexual modesty and be chaste, otherwise they will be exposed to ridicule and shame [17]. Similarly, gender norms dictate that women should be submissive. The rising number of female students in universities has not kept pace with the entrenched gender expectations [10, 17]. Women are still unable to reach higher positions as some men still think that women should remain at home and take care of their families [17, 33].

In Sri Lankan society where patriarchal practices dominate, 'power over' is the currency used to gain social dominance and respect. Study participants felt that ragging was an opportunity for seniors to form hierarchies and therefore hold power over the newcomers. Participants reported it was mandatory for the juniors to obey the seniors. Furthermore, the former newcomers looked forward to their turn as seniors, gaining control over the next new batch, maintaining the cycle of ragging [17]. The requisite for deference towards parents, authoritative figures and one's seniors is the cultural norm [34]. Senior students by default acquire this position of power, and thereby pacifying their frustrations over social inequalities, personal jealousies and inferiority complexes [10, 11, 14]. Worldwide studies show that coercion, domination and abuse of power are the pillars that these initiation rites are built upon [7, 8, 35].

Ragging also invokes Sri Lankan society's acceptable masculinity role. This study supports the notion that ragging among men often takes a more aggressive and physical form as compared to the more psychological form women endure. Correspondingly, several studies found these practices to be gendered [12, 36]. Véliz-Calderón et al. [36] and Tong et al. [37] describe hazing experiences among female students to be psychological, including eating disgusting food and sleep deprivation, whereas male students had to undergo activities displaying physical strength, supporting the socio-cultural construct of American masculinity/femininity.

Evidence of gendered male dominance and female submission was also present in our study. These gendered expectations are damaging to both men and women. Attracting a partner is one such example [10, 17]. Senior male students were expected to demonstrate power over the newcomers which enabled them to impress a suitable partner from the opposite sex to fulfill their romantic and sexual needs [11, 17]. Due to the notion of '*Læjja-baya*' expressed earlier, male interaction with females is limited and ragging provides a platform for connecting.

With social dominance and the quest for power at stake, it is therefore not surprising that ragging is maintained in secrecy. This has a historical implication where ragging has only recently been discussed. Despite numerous efforts by the UGC and the Sri Lankan government, ragging continues to be widespread. Like ragging, hazing and other initiation practices are often secretly conducted according to Campo et al. [38]. This secrecy is in part owed to the unwillingness of students to make complaints due to the fear of being ostracized and of the wrath of powerful seniors [13]. This study's participants expressed similar fears that were outweighed by the authorities' apathy they have experienced upon reporting. Gunatilaka et al. [10] reported ragging was seen by students, as a small price to pay to receive a university education and all that entails, also seen in this study. Participants claimed refusal to participate in ragging resulted in a loss of inclusion. The practice of excluding or "othering" is an instinctive reaction to protect oneself and one's group particularly when there is a perception of scarce resources [39]. This holds true when many students are fighting for limited university seats and employment opportunities.

Students in this study often normalized or trivialized violence. The dismissive manner in which violence is seen by society can play a role in the acceptability of ragging. Sri Lanka has had a violent past with several bloody insurgencies and a protracted ethnic uprising [18].These study participants were born during Sri Lanka's days of civil war. This perceived invisibility/ normalization of violence has also permeated the culture in the ways children are raised. This could have influenced the tolerance of violent and aggressive behavior [10]. Although corporal punishment in schools is officially banned, it continues to occur at home [40]. Child maltreatment contributes to a child's normalization of violence according to studies conducted globally [41, 42]. Several other studies on intimate partner violence in Sri Lanka have shown violence rates to be between 17–72% [43, 44], these permissive attitudes towards violence against women also demonstrate societal perceptions of violence [10]. Normalization of violence distorts ragging as harmless. This has been described in other studies where students don't acknowledge these practices as harmful or deny having experienced what they consider to be 'violence' [7, 38].

Another aspect contributing to student ragging is the rapid expansion of government funded universities and the influx of diverse student populations that came along with this. It did not go hand in hand with sufficient infrastructure, accommodation, leisure activities and sports for students [18]. This could be attributed to slow growth of the economy and the lack of public funds to finance educational institutes [14]. The scarcity of funding has also resulted in shortcomings such as a lack of lecturers, decreased quality of teaching and poor security within the university and surroundings [13]. The participants indicated that these unsatisfactory conditions and inadequacies at the system level contribute to student rebellion, which is in line with Galtung's structural violence [25]. Societal structures that honor certain groups and not others, contribute to the invisible violence level, in this case, manifested as ragging.

## The spectrum of ragging

The students' ambivalence revealed that ragging is seen on a spectrum with both positive and negative attributes. Students who saw ragging as more of a bonding experience, were proponents for ragging; they experienced it as a part of university life and a rite of passage all new entrants must undergo in order to belong [17]. Ragging can range from performing trivial tasks such as singing songs to extreme physical and sexual harassment [10]. Other studies also showed that these practices increase group cohesiveness [45, 46]. Participants claimed that certain ragging practices were constructive and helpful, especially milder forms have been shown to have positive effects on students similar to other studies [10]. Students entering university

from rural areas or disadvantaged backgrounds could be timid, making it difficult for them to engage and communicate with other students and lecturers. By performing simple ragging tasks helping them overcome shyness and experience a sense of belonging [17]. Due to the cited shortcomings or absence of university administration support, newcomers felt more reliant upon senior students for support/guidance.

The participants of our study mentioned several negative effects of ragging, which were similar to findings in other studies that show severe forms of ragging result in adverse consequences for their victims, such as physical and emotional problems [20, 47]. Ragging disrupts the education of the 1st year students since it mostly occurs during this period. Similarly, students have recounted in an UGC report, to be unable to concentrate due to stress and sleep deprivation [48] leading to poor exam performance, delays in graduating and entering the job market [13]. The negative impression generated by ragging in state-funded universities, has been shown to impede the chances of graduates gaining employment in the private sector [32].

### Strengths and limitations

There were several limitations to our study. Ragging victimization mostly occurs in the 1st year and since the participants were 2nd and 3rd years students there may have been a recall bias as they could have been victims, perpetrators or by-standers as well as affected by the cultural tendency to trivialize/normalize ragging. As this is a sensitive topic and victims can become future perpetrators, they may have refrained from revealing information.

The FGDs were conducted after a severe ragging incident in the Technology faculty which led to the closure of this faculty for several weeks which could have made the students more reluctant to discuss ragging. There is the possibility that students provided what they assumed to be desirable answers, though our experience was that they were eventually quite forthright about positive and negative issues.

The research team did have only female members; this could have affected the information divulged by male students. It would be difficult to say if males would have been more or less comfortable sharing females considering the gendered expectations upheld in society.

The main strength of our study was that our findings focused upon students' perspectives. Assisted through the chosen theoretical lenses, a more nuanced view of students' behaviour was illuminated. Most existing information on ragging in Sri Lanka is derived from reports and studies that lack a rigorous research methodology. This study is at the forefront of including students in the analysis of this very sensitive and multi-layered social problem.

Further research is needed to better understand the practice of ragging. Replicating this methodology focusing on student voices is essential, as there is a dearth of research concerning students' perspective. As in most qualitative studies, transferability to other settings is possible but must be done with caution as context plays such an important role. Participatory action research could be useful in providing agency to those implicated on the front lines of ragging. The perceptions of lecturers and key persons in positions of authority in the university as well as parents' perceptions would also enhance a more comprehensive picture of ragging.

### Conclusion

This study suggests students use ragging as an expression of power in a highly competitive societal context which includes the educational system. Some of the contributing factors towards this practice are normalization of violence, the acceptance of ragging as part of the university subculture, gender norms and socioeconomic class disparities. These factors, when combined, provides a breeding ground for student unrest, violence and insubordination, manifesting as ragging. This study has illuminated a wider picture by providing students' insights,

which are a fundamental part of a multisectoral approach towards solutions; one that involves university administration, the UGC, student unions and university students themselves. A proactive approach is needed; one that creates awareness about ragging's harmful effects, while promoting a more positive interaction between students and those in positions of authority. This long-standing problem has reached a critical juncture of doing more harm than good to young people while acquiring education and developing skills for life, so important to the future of their country. Therefore, all efforts must be used to eliminate ragging and its deleterious effects. There is a serious need for deep reforms within universities and a critical look at the role of structural violence, to successfully address ragging.

## Supporting information

**S1 File. Focus group discussion interview guide.**
(PDF)

**S2 File. COREQ (Consolidated criteria for Reporting Qualitative research) checklist.**
(PDF)

**S3 File. Types of ragging and incidents of ragging.**
(PDF)

**S4 File. Transcripts of FGDs.**
(PDF)

## Acknowledgments

Rajendra Surenthirukumaran, Kumudu Wijewardana, Shajeetha Thurauappah are thanked for their time and effort in making this study possible.

## Author Contributions

**Conceptualization:** Ayanthi Wickramasinghe, Pia Axemo.

**Data curation:** Ayanthi Wickramasinghe, Pia Axemo.

**Formal analysis:** Ayanthi Wickramasinghe, Pia Axemo, Jill Trenholm.

**Funding acquisition:** Pia Axemo, Birgitta Essén.

**Investigation:** Ayanthi Wickramasinghe, Pia Axemo.

**Methodology:** Ayanthi Wickramasinghe, Pia Axemo, Birgitta Essén, Jill Trenholm.

**Project administration:** Ayanthi Wickramasinghe, Pia Axemo.

**Resources:** Ayanthi Wickramasinghe, Pia Axemo, Jill Trenholm.

**Software:** Ayanthi Wickramasinghe.

**Supervision:** Pia Axemo, Birgitta Essén, Jill Trenholm.

**Validation:** Ayanthi Wickramasinghe, Pia Axemo, Birgitta Essén, Jill Trenholm.

**Visualization:** Ayanthi Wickramasinghe.

**Writing – original draft:** Ayanthi Wickramasinghe.

**Writing – review & editing:** Ayanthi Wickramasinghe, Pia Axemo, Birgitta Essén, Jill Trenholm.

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
