## [Decision Letter · Decision Letter 0]

10 Aug 2021

PONE-D-21-15111

“We are considered to be dust”; a qualitative study on students perceptions on the phenomenon of ragging at a Sri Lankan university

PLOS ONE

Dear Authors,

Thank you for submitting your manuscript to PLOS ONE. After careful consideration, we feel that it has merit but does not fully meet PLOS ONE’s publication criteria as it currently stands. Therefore, we invite you to submit a revised version of the manuscript that addresses the points raised during the review process.

We look forward to receiving your revised manuscript.

Kind regards,

Marcel Pikhart

Academic Editor

PLOS ONE

Journal Requirements:

2. Please include additional information regarding the interview guide used in the study and ensure that you have provided sufficient details that others could replicate the analyses. For instance, if you developed an interview guide as part of this study and it is not under a copyright more restrictive than CC-BY, please include a copy, in both the original language and English, as Supporting Information.

Reviewers' comments:

Reviewer's Responses to Questions

**Comments to the Author**

1. Is the manuscript technically sound, and do the data support the conclusions?

Reviewer #1: Partly

Reviewer #2: Partly

Reviewer #3: No

2. Has the statistical analysis been performed appropriately and rigorously? 

Reviewer #1: N/A

Reviewer #2: Yes

Reviewer #3: No

3. Have the authors made all data underlying the findings in their manuscript fully available?

Reviewer #1: Yes

Reviewer #2: Yes

Reviewer #3: No

4. Is the manuscript presented in an intelligible fashion and written in standard English?

Reviewer #1: Yes

Reviewer #2: Yes

Reviewer #3: Yes

5. Review Comments to the Author

Reviewer #1: Dear Authors,

This manuscript explores students’ perceptions on the phenomenon of ragging in a university. Any type of bullying, without a doubt, constitutes a serious problem that must be studied in depth.

This is an interesting and important study due, despite this, before it can be published it would need some methodological clarifications and revisions:

- The manuscript does not meet with the sections of the journal.

- Introduction/background seem very long and confused. This journal only has an introductory section. This structure looks like for another journal. The claims are not properly placed in the context of the previous literature, since there are references that show contexts or social situations from more than 20 years ago.

Materials and methods section: “Qualitative research is an approach for exploring and understanding the meaning individuals or groups ascribe to a social or human problem” (Creswell, 2013b, p. 3). In this case, the theoretical approach has not been indicated, is it ethnography, phenomenology, grounded theory? A correct qualitative study needs a strong theoretical foundation.

- Research design needs to be further explained, as well as other subsections of the methods. Following the journal recommendations, I kindly suggest reviewing the CASP and COREQ checklists to ensure the quality of the data reporting. For example, the researcher's relationship with the participants, What grade and semester were the students in?

- I suggest that only current references are used in the discussion, due to changes in this phenomenon in recent years (for example, the introduction of new technologies has changed the type of harassment in different contexts).

- You should mention limitations.

Thank you for the clarifications.

Reviewer #2: I want to thank you for the opportunity to review this manuscript. The time spent creating and shipping it is greatly appreciated. In my humble opinion, it offers interesting results that the scientific community, as well as professionals in this field, can benefit from. However, currently the manuscript presents some problems that must be taken into account and repaired. Here are my recommendations:

There is a total absence of contextualization of the study, the introduction does not justify or theoretically support the research, they should include the hypotheses of the work.

The method is not explicit enough. It does not give details of the data analysis and the software used for the analysis of the transcripts. Likewise, you should expand the information on the characteristics of the participants.

In the discussion section there is no comparison of its results with the initial contextualization. Personally, I am not in favor of using literature in the discussion that has not been previously mentioned in the introduction, since it transmits an incomplete review of the state of the topic investigated, there are several of the citations included in this section that have not been previously cited in the introduction .

It should include the limitations and future lines of research before the conclusions.

The conclusions can be expanded to include and / or give examples of how this information can be useful to the population and the field of education. Authors must make explicit reference to the practical application of the results obtained (this is one of the strengths of the manuscript), paying special attention to the possibilities offered by the data for the design of interventions.

These limitations should be addressed with a view to the possible publication of the manuscript.

Thank you for your work.

Reviewer #3: This is an intriguing report of the findings emerging from focus groups at a university.

Introduction/Background/Theoretical Lenses:

- This section needs more of the explanatory material provided in the Discussion. Please use the introduction to expand more on issues of how ragging is perhaps culturally specific, even though it does have analogues in other cultures. Indeed, please move up much of the information on Sri Lankan society, especially the social classes, that is given in the Discussion. Those facts help the reader to make sense of the data you report and you should give more of that information in your Introduction to help the reader contextualize the findings. You don't need to to talk about all of the details in the introduction, but they do add information that would be helpful to a reader of the introduction

Results:

- The thematic analysis is interesting, but the reader has no sense of how representative it is of the sample of focus groups or individuals. Please go back to the data and make tables to report the number of times each of these themes were mentioned or discussed, variation by gender, and ethnicity. You can also use such data to conduct a grander analysis, like a cluster analysis for the themes. The reader is curious, does the expression of some of these themes within the focus groups overlap with the expression of other themes? That is, are the themes correlated? If so, you could report and discuss such potential patterns in the data. At present, the reader has a rich picture of the general findings and concept of ragging and its social-economic expression at the University, but more concrete numbers would greatly enhance the paper.

Conclusion:

- Please reduce the space given to the Discussion (currently 6 pages) and insert much of that important information in the Background/Theoretical Lenses (which are only 3 pages combined)

Thank you for conducting this important research.

6. PLOS authors have the option to publish the peer review history of their article (what does this mean?). If published, this will include your full peer review and any attached files.

Reviewer #1: No

Reviewer #2: No

Reviewer #3: No

---

## [Author Response · Author response to Decision Letter 0]

3 Sep 2021

We appreciate the time and effort that you and the reviewers have dedicated to providing your valuable feedback on our manuscript. We are grateful to the reviewers for their insightful comments on our manuscript. We have been able to incorporate changes to reflect most of the suggestions provided by the reviewers. We have highlighted the changes within the manuscript. Please see below for a point-by-point response to the reviewers’ comments and concerns. All page numbers refer to the revised manuscript file with tracked changes. 

Reviewer 1#

Thank you very much for taking the time to assess our manuscript and provide your valuable comments. We have addressed all the concerns you have raised to the best of our abilities.

- The manuscript does not meet with the sections of the journal.

We thank the reviewer for pointing this out. We have revised the manuscript accordingly to comply with the journal format throughout the manuscript. 

We have also changed the heading Background to Introduction (page 3, line 46). 

- Introduction/background seem very long and confused. This journal only has an introductory section. This structure looks like for another journal. The claims are not properly placed in the context of the previous literature, since there are references that show contexts or social situations from more than 20 years ago.

We have changed the word background to introduction along with a few other formatting changes as mentioned above. 

We have changed the introduction (pages 3-5) to provide a better contextual background as we also received a similar comment from another reviewer. 

We agree with the reviewer that some references date back 20 years but there is a strong cultural and historical context to the occurrence of ragging that we wished to highlight. At the same time this is an under researched area in Sri Lanka and there is a lack of current research that can portray the social situation and context. 

- Materials and methods section: “Qualitative research is an approach for exploring and understanding the meaning individuals or groups ascribe to a social or human problem” (Creswell, 2013b, p. 3). In this case, the theoretical approach has not been indicated, is it ethnography, phenomenology, grounded theory? A correct qualitative study needs a strong theoretical foundation.

Thank you for pointing out our oversight in not mentioning the theoretical approach. We have now included it in (page 7, line 198). Our reference for the phenomenological approach aligns with Creswell that states that, “Qualitative research is an approach for exploring and understanding the meaning individuals or groups ascribe to a social or human problem”.

We agree that qualitative studies need a strong theoretical foundation and you will find our theoretical framework in figure 1. 

- Research design needs to be further explained, as well as other subsections of the methods. Following the journal recommendations, I kindly suggest reviewing the CASP and COREQ checklists to ensure the quality of the data reporting. For example, the researcher's relationship with the participants, what grade and semester were the students in?

That was a very good suggestion to align with the COREQ checklist. We have revised our methods and materials section using the COREQ checklist. The changes are visible on (pages 8-10, lines 216-278).

The researcher had no relationship with the participants and further details have been added to the relevant section (page 8, line 220-228). To maintain the anonymity of the participants, we did not inquire which grade or semester they were currently studying-in but all participants were chosen from the 2nd and 3rd year of study (page 8, line 217-220). 

- I suggest that only current references are used in the discussion, due to changes in this phenomenon in recent years (for example, the introduction of new technologies has changed the type of harassment in different contexts).

We appreciate the reviewer’s suggestion and agree that it would be better to use more current references, unfortunately since there is little research available, we are unable to do so. We share your belief that technology has enhanced the ability to be bullied and there were instances stated by our participants where mobile phones were used. It is our hope that this seminal paper will lead to deeper and broader inquiries into the myriad of ways that ragging is carried out in Sri Lanka. The public health of youth and particularly the increase in mental health issues, demand this.

- You should mention limitations

Thank you for pointing out this omission. We have rectified this mistake and added the limitations of our study (page 32, lines 929-940). 

Reviewer 2#

We appreciate your time and effort in providing feedback on our manuscript and are thankful for the insightful comments. We have tried our best to address all your comments and concerns. 

- There is a total absence of contextualization of the study, the introduction does not justify or theoretically support the research, they should include the hypotheses of the work.

Thank you for this feedback concerning the lack of context in the introduction. We have rectified that shortcoming by reorganizing the introduction section and adding more context as per the reviewer’s suggestion. These changes are reflected in the revised introduction section (pages 3-5). We appreciate the suggestion to include a hypothesis, however as this manuscript is an explorative qualitative study, as in most studies of this nature, there is rarely a hypotheses as it is more about discovery. 

- The method is not explicit enough. It does not give details of the data analysis and the software used for the analysis of the transcripts. Likewise, you should expand the information on the characteristics of the participants.

We agree with the reviewer and have updated the methods section to provide more details and information. These changes can be seen on (page 8, lines 214-220 and page 8, lines 253-278). No software was used for the analysis of the transcripts. The diverse research team read and re-read the transcripts after which we spent several days altogether mapping, defining, and re-defining the themes with the help of a whiteboard as per Braun and Clark’s description of carrying out thematic analysis. 

- In the discussion section there is no comparison of its results with the initial contextualization. Personally, I am not in favor of using literature in the discussion that has not been previously mentioned in the introduction, since it transmits an incomplete review of the state of the topic investigated, there are several of the citations included in this section that have not been previously cited in the introduction.

Thank you for this observation. We have changed the introduction (pages 3-5) to improve the context as well as changed the discussion (pages 26-31), thereby allowing a better comparison of the results in the discussion section. In this explorative study design, new findings emerged that required searching for new literature that was then included in the discussion. Prior to analysis of the findings, this new material was not relevant to the introduction. 

- It should include the limitations and future lines of research before the conclusions.

The conclusions can be expanded to include and / or give examples of how this information can be useful to the population and the field of education. Authors must make explicit reference to the practical application of the results obtained (this is one of the strengths of the manuscript), paying special attention to the possibilities offered by the data for the design of interventions. These limitations should be addressed with a view to the possible publication of the manuscript.

- We agree that practical application is our eventual goal. We have strengthened the manuscript to include the reviewers suggestions, which will be reflected on (pages 32-33, lines 942-949). As this was an explorative study, these results cannot be generalized although in qualitative there is a possibility that some of the findings/ideas are transferable to similar settings/contexts but that would be up to the different individual readers to draw any conclusions there. Therefore, the practical application of the results and the design of intervention require further research. This will certainly be addressed in our upcoming research currently underway.

The limitations of our study have been included to the manuscript 

(page 32, lines 929-940).

Reviewer 3#

We thank you for taking the time to identify certain issues in our manuscript and providing us with the opportunity to strengthen the paper. 

- Introduction/Background/Theoretical Lenses:

This section needs more of the explanatory material provided in the Discussion. Please use the introduction to expand more on issues of how ragging is perhaps culturally specific, even though it does have analogues in other cultures. Indeed, please move up much of the information on Sri Lankan society, especially the social classes, that is given in the Discussion. Those facts help the reader to make sense of the data you report and you should give more of that information in your Introduction to help the reader contextualize the findings. You don't need to talk about all of the details in the introduction, but they do add information that would be helpful to a reader of the introduction.

Thank you for bringing this to our attention what lacked in our introduction to make this paper a comprehensive read for those unfamiliar with this context. We have reorganized and revised our introduction (pages 3-5) and discussion (pages 26-31) as suggested to include more extensive information concerning Sri Lanka’s history, society, and culture. 

- Results:

The thematic analysis is interesting, but the reader has no sense of how representative it is of the sample of focus groups or individuals. Please go back to the data and make tables to report the number of times each of these themes were mentioned or discussed, variation by gender, and ethnicity. You can also use such data to conduct a grander analysis, like a cluster analysis for the themes. The reader is curious, does the expression of some of these themes within the focus groups overlap with the expression of other themes? That is, are the themes correlated? If so, you could report and discuss such potential patterns in the data. At present, the reader has a rich picture of the general findings and concept of ragging and its social-economic expression at the University, but more concrete numbers would greatly enhance the paper.

We appreciate the reviewer’s suggestion and agree that it would be interesting to demonstrate number of times each of these themes were mentioned or discussed, variation by gender, and ethnicity however, such an analysis is beyond the scope of this introductory and exploratory paper. We hope that this explorative/scoping study will serve as base line information to feed into our planned quantitative study where your idea a grander analysis would indeed provide more concrete numbers. 

The themes of our study were inductive and data driven. In the method of Thematic analysis as per Braun and Clark, themes can overlap and are not mutually exclusive as in content analysis. This is demonstrated in figure 2. 

- Conclusion:

Please reduce the space given to the Discussion (currently 6 pages) and insert much of that important information in the Background/Theoretical Lenses (which are only 3 pages combined). 

Thank you for your comment, we have enriched the introduction and reorganized the discussion section as per the reviewer’s suggestion. Unfortunately, we were not able to reduce the number of pages of the discussion since we had to address the comments from the other reviewers on the discussion, and we also added the limitations of our study.

---

## [Decision Letter · Decision Letter 1]

20 Sep 2021

PONE-D-21-15111R1

“We are considered to be dust”; a qualitative study on students perceptions on the phenomenon of ragging at a Sri Lankan university

PLOS ONE

Dear Authors,

Thank you for submitting your manuscript to PLOS ONE. After careful consideration, we have decided that your manuscript does not meet our criteria for publication and must therefore be rejected based on the reviwers´ suggestion. 

I am sorry that we cannot be more positive on this occasion, but hope that you appreciate the reasons for this decision.

Yours sincerely,

Marcel Pikhart

Academic Editor

PLOS ONE

Reviewers' comments:

Reviewer's Responses to Questions

**Comments to the Author**

1. If the authors have adequately addressed your comments raised in a previous round of review and you feel that this manuscript is now acceptable for publication, you may indicate that here to bypass the “Comments to the Author” section, enter your conflict of interest statement in the “Confidential to Editor” section, and submit your "Accept" recommendation.

Reviewer #1: (No Response)

Reviewer #2: All comments have been addressed

Reviewer #3: All comments have been addressed

2. Is the manuscript technically sound, and do the data support the conclusions?

Reviewer #1: Partly

Reviewer #2: Yes

Reviewer #3: Partly

3. Has the statistical analysis been performed appropriately and rigorously? 

Reviewer #1: N/A

Reviewer #2: Yes

Reviewer #3: No

4. Have the authors made all data underlying the findings in their manuscript fully available?

Reviewer #1: No

Reviewer #2: Yes

Reviewer #3: No

5. Is the manuscript presented in an intelligible fashion and written in standard English?

Reviewer #1: No

Reviewer #2: Yes

Reviewer #3: Yes

6. Review Comments to the Author

Reviewer #1: (No Response)

Reviewer #2: (No Response)

Reviewer #3: Thank you for addressing the concerns of my review. The paper is now much easier to follow for a non-specialist reader. However I do remain concerned about the lack of any type of quantitative analysis of the data, but I understand that such analyses will be provided in future research.

7. PLOS authors have the option to publish the peer review history of their article (what does this mean?). If published, this will include your full peer review and any attached files.

Reviewer #1: No

Reviewer #2: No

Reviewer #3: No

- - - - -

---

## [Author Response · Author response to Decision Letter 1]

6 Dec 2021

Comments by the Author

1. If the authors have adequately addressed your comments raised in a previous round of review and you feel that this manuscript is now acceptable for publication, you may indicate that here to bypass the “Comments to the Author” section, enter your conflict of interest statement in the “Confidential to Editor” section, and submit your "Accept" recommendation.

Reviewer #1: (No Response)

Reviewer #2: All comments have been addressed

Reviewer #3: All comments have been addressed

Authors comments

Based on the question it appears that two reviewers were satisfied with our comments which is why we are confused where our omissions lie. 

2. Is the manuscript technically sound, and do the data support the conclusions?

Reviewer #1: Partly

Reviewer #2: Yes

Reviewer #3: Partly

Authors comments

This question is very quantitatively oriented and is not applicable to our manuscript as it was a qualitative manuscript. Our manuscript is technically sound for a qualitative piece, the data supports the conclusions and has been guided by the COREQ checklist suggested by your reviewer. Our manuscript also follows the guidelines for qualitative work on the PLOS ONE website submission guide. 

3. Has the statistical analysis been performed appropriately and rigorously? 

Reviewer #1: N/A

Reviewer #2: Yes

Reviewer #3: No

Authors comments

This question is also quantitatively oriented and is not applicable to our manuscript as it was a qualitative manuscript. We are quite surprised by these varied responses by your reviewers number 2 and 3. 

4. Have the authors made all data underlying the findings in their manuscript fully available?

Reviewer #1: No

Reviewer #2: Yes

Reviewer #3: No

Authors comments

We were asked to provide the interview guide which we promptly provided. There were no further requests for any other data although we would have gladly provided any further data if requested. 

In review 2, we received contradictory responses from the reviewers as opposed to review 1, which leaves us quite confused. 

5. Is the manuscript presented in an intelligible fashion and written in standard English?

Reviewer #1: No

Reviewer #2: Yes

Reviewer #3: Yes

Authors comments

We are quite perplexed that in the first review round, all reviewers agreed, the manuscript was “presented in an intelligible fashion and written in standard English”. In the second-round reviewer number 1 changed the response although only minor edits were carried out according to the reviewers comments. The manuscript has undergone English language review each time by a professional English academic writer, who is also a co-author. 

6. Review Comments to the Author

Reviewer #1: (No Response)

Reviewer #2: (No Response)

Reviewer #3: Thank you for addressing the concerns of my review. The paper is now much easier to follow for a non-specialist reader. However I do remain concerned about the lack of any type of quantitative analysis of the data, but I understand that such analyses will be provided in future research.

Authors comments

If our manuscript is “not acceptable” despite “all comments have been addressed” (please see comment 1), it leaves us wondering what is missing now and leaves little room for improving the paper, qualitatively. 

The comment from reviewer 3 shows lack of understanding of qualitative work as there is no quantitative analyses in qualitative research. The reviewer #3 also presents as a “non-specialist reader”.

---

## [Editor Report · Decision Letter 2]

22 Feb 2022

PONE-D-21-15111R2

“We are considered to be dust”; a qualitative study on students perceptions on the phenomenon of ragging at a Sri Lankan university

Dear Dr Wickramasinghe,

First of all, let me thank you for your patience! I have now finished my assessment of your paper and the previous review process.

Your paper touches on a very important issue that is worth to be presented to the scientific public. However, I have a couple of major concerns that prevent the publication of the paper as it stands. If you see a possibility and feel motivated for a second major and thorough revision that addresses my concerns, it will be a pleasure for us to review this revised version. But I should emphasize that it will be an open decision and that there will be no guarantee that PlosOne publishes a revised version.

Before I go into details of the paper, I will explain my assessment of the previous review process. Similar as you, I was surprised by the rejection of the paper on the basis of three very different reviewer reports. Although it is not unusual that the paper is rejected even if one reviewer accepts the paper, another one wishes major revision and the third reviewer rejects the paper, I think you were justified to expect more involvement and guidance from the academic editor, particularly because the reviewer recommendations diverged. I also share your opinion that at least one of the reviewer may have been a suboptimal choice since s(he) was not very familiar with qualitative methods and asked you to present quantitative details that are only usual in some qualitative methods, such as a qualitative content analysis.

Therefore, I think your appeal was justified as far as we consider the review process. Together with a Division Editor of PlosOne, we decided to re-review your paper. To avoid an undue delay, only one person, me as the new Academic Editor, has reviewed the paper:

1. My first major concern refers to the study question. Although the study question  seems clear on first view – the perception of ragging, seen from the student perspective –, the sample of students that you choose for your study makes the issue ambiguous. One could say, it is a group ‘in - between’. They are no longer freshman who experience ragging at the very moment; they are not 'old' students who are 'entitled' to rag other students. Therefore, some of them may have experienced ragging – we don’t know exactly; some may have observed ragging being in the position of a 'neutral observer' or a 'former affected person' – we don’t know exactly; some seem to prepare themselves to become ‘ragger’ themselves in the near future – we don’t know exactly. But throughout the paper you write more or less as an advocate of the 'victims' (see, for example, lines 114 -118).

I can understand your argument that you did not want to discuss this matter with young students, who may still suffer from ragging or are reminded of this terrible experience in the group discussion, you cannot claim your study addresses the victim perspective (line 116). While this seems to be a limitation of your study, you could also make it to a strength of your study. The experiences of your 'in – between' group may be an optimal opportunity to study how a procedure, which is so burdensome for many young persons, becomes an established and accepted institution in students' life. So you should clarify what you exactly expect from your study – it is definitely not the experience of ragging; in this case young students should have been enrolled. So you should precisely define your scientific interest (e.g., to understand how ragging invades student life and culture). Of course, you can criticize the practice of ragging in the Discussion but you should not use the study participants for this purpose; they are much more ambivalent.

2. My second major concern refers to what you define as the 'overarching theme' of your analysis: "ragging as a means of communication". It was you, not the students, who stimulated a communication about ragging. Even more important, the students first reaction in the group discussion was to emphasize that ragging did not occur (line 391). It seems to be a secret or hidden practice, but definitely not a ‘means of communication’. Perhaps you mean something else but your expression is in any case misleading. By the way, I’m not sure whether you need an overarching theme at all. Isn’t it sufficient to define your subthemes as main themes and to present them as your results?

In this case, it may be a good idea to start the Results section with the issue of 'secrecy' and your overall observation that all groups denied ragging in the beginning of the discussion. This is really striking, given the formative experience of ragging. Obviously, older students try to forget or deny it. This could be the first important result and could later lead to an interesting discussion how difficult it is to make ragging 'public'.

3. Please check the logic of your themes and how you present them in the Results section. I see a couple of inconsistencies so that a re-arrangement may be required. Let me give you several examples:

(a) ‘Trivializing violence’ (lines 415 ff.) is not a subtheme of "Veil of secrecy and silence".

(b) Although the examples under the subtheme "Rigid norms" are interesting, I’m not sure whether all of them belong to the main theme "A society with deep divisions". They deal, for example, with things like a general lack of protection of women (why is this a matter of ‘rigid norms’?). I think, other interesting examples, too, have nothing to do with ‘rigid norms’, such as equalizing (lines 503 ff.) or denying ethnic clashes (lines 536 ff.). Please check the main theme and the subthemes so that they are consistent.    

(c) It is also not clear how the students’ feeling not to be heard by the society and the university (lines 542 ff.) or the feeling of insecurity (lines 571 ff.) is related to ragging. It is surely an important aspect, especially in the life of minorities but the authors should help readers to understand why these feelings and experiences may lead to ragging or may cause some students to accept ragging or to deny it. This should happen at least in the Discussion

(d) I would recommend to display “The students’ recommendations” (lines 585 ff.) as a main theme in Figure 2 and to present it as a main theme in the text so that readers can better follow your complete analysis and identification of themes.

4. The Discussion needs much more structure. You should start with a short and precise “summary of main findings”, no longer than 8 to 10 lines. I would suggest to stress the aspect of ambiguity or ambivalence towards ragging as the most important aspect (at least for me). The first sentence (lines 611 f.) Is superfluous; the last sentence (lines 617 ff. should be moved to the "Strength and Limitations" section in the Discussion. After the short summary you could go on with three major sections perhaps clearly marked with a subheading:

- Why ragging is positively perceived?

- Why ragging is negatively perceived?

- The wider context of ragging: Sri Lanka and its university culture

(this is nothing nothing more than a suggestion; I think you have better ideas.)

In any case, it is important that the reader recognizes a clear structure and a clear line of arguments.

5. I like your theoretical model of violence displayed in Figure 1, but I see two problems. I miss an “integration” as you claim with the Figure. Some arrows in the Figure and the pure statement “structural violence, intersectionality and social dominance theories are intricately linked” in the text (line 144) are not an integration. Moreover, you should use this model in a more creative way to explain ragging in societies like Sri Lanka. For example, as far as I understand, you say Sri Lanka is a violent country and unsatisfactory conditions contribute to student violence (line 657 f.). I agree, but one could ask, how? You say, ragging is violence and at the same time, ragging seems to be protest against inequalities and violence of the society. It is also remarkable that it is not the upper-class students (or only a few) who practice ragging but those at the other social end. Therefore, I would very much appreciate if you make a second attempt to use the model for an explanation.

Some minor concerns:

6. In addition to reference 22, you may also refer to this just published study that shows the strong association of ragging and suicide: https://pubmed.ncbi.nlm.nih.gov/35025905/

7. Did you attach the COREQ statement for qualitative studies and did you fill in the page numbers, as requested?

8. Line 399: One or two examples how seniors named juniors could be helpful to better understand this humiliating procedure.

9. It would be generally helpful to give more examples (in form of an Appendix) how ragging is executed. It reminded me of practices and rituals of student leagues (“Burschenschaften) in Germany. Here, too, the younger ones have to be servants of the older students, are treated very badly, much alcohol is on the way and for some student leagues it is a special pleasure to hurt one another, using swords and, thus, to demonstrate masculinity. These practices are well known and also documented via media so that we have a very good idea of these rituals. Perhaps you have also some material to present examples in an Appendix that makes ragging easier imaginable for Non-Sri Lanka people.

10. Line 729 ff. Of course, the practices you describe here are humiliating women. But please, consider that it can also be a hard expectation towards male students to behave in such an inhuman way. I do not want to exonerate males from accusations but some of them may also feel in a dilemma.

11. Line 785: I agree with your last plea but, following the students’ experiences, should it not read that it needs deep reforms in the university and the society to successfully combating ragging?

We look forward to receiving your revised manuscript.

Kind regards,

Wolfgang Himmel

Academic Editor

PLOS ONE

1. We notice that your manuscript file was uploaded on September 3, 2021. Please can you upload the latest version of your revised manuscript as the main article file, ensuring that does not contain any tracked changes or highlighting. This will be used in the production process if your manuscript is accepted. Please follow this link for more information: http://blogs.PLOS.org/everyone/2011/05/10/how-to-submit-your-revised-manuscript/

---

## [Author Response · Author response to Decision Letter 2]

4 Apr 2022

Response to Reviewers 

Dear Wolfgang Himmel, 

Thank you for your review of our manuscript. We appreciate the time and effort you have taken to provide insightful feedback as well as your useful comments. Your comments and concerns have been addressed below. We accept the opportunity you have offered to submit a revised draft of our manuscript with the revised title: “When a person can’t obey, he is subjected to violence”; a qualitative study on students perceptions on the phenomenon of ragging at a Sri Lankan university to PLOS ONE. Attached, please also find the completed COREQ guide as requested and additionally, more examples of ragging, as supporting information in an Appendix. 

The interview guide was developed by the research group as a part of the study and is not copy righted. The interview guide was only developed in English as all the moderators were fluent in English and did not require the interview guide to be translated. 

We attached the transcripts of the 17 FGDs as requested as supporting information S4_4.File. We would once again like to point out the sensitive nature of the transcripts. The FDGs conducted among the students were divided by ethnicity and gender, and most often the participants belonged to the same faculty. During the FGDs the participants also discussed certain issues which would increase the chances of them being identified. Some of the issues discussed where regarding the lecturers and the university administration which could cause problems for the students if they were to be identified. These indirect identifiers may risk the identification of study participants, which would be a breach of our confidentiality agreement, as well as our ethical approval. 

Changes made within the manuscript are highlighted below in blue along with a point-by-point response to your comments and concerns. All page numbers refer to the revised manuscript file with tracked changes. 

1. My first major concern refers to the study question. Although the study question seems clear on first view – the perception of ragging, seen from the student perspective –, the sample of students that you choose for your study makes the issue ambiguous. One could say, it is a group ‘in - between’. They are no longer freshman who experience ragging at the very moment; they are not 'old' students who are 'entitled' to rag other students. Therefore, some of them may have experienced ragging – we don’t know exactly; some may have observed ragging being in the position of a 'neutral observer' or a 'former affected person' – we don’t know exactly; some seem to prepare themselves to become ‘ragger’ themselves in the near future – we don’t know exactly. But throughout the paper you write more or less as an advocate of the 'victims' (see, for example, lines 114 -118).

Thank you for pointing this out. We have revised the manuscript accordingly to make it clearer. As health care advocates we often tend to divert attentions towards the victim’s plight but we agree that this is a public health problem where all students suffer, as you correctly indicate. 

These changes are visible on page 8 line 223-224 and in the introduction in page 6, line 136-138.

I can understand your argument that you did not want to discuss this matter with young students, who may still suffer from ragging or are reminded of this terrible experience in the group discussion, you cannot claim your study addresses the victim perspective (line 116). While this seems to be a limitation of your study, you could also make it to a strength of your study. The experiences of your 'in – between' group may be an optimal opportunity to study how a procedure, which is so burdensome for many young persons, becomes an established and accepted institution in students' life. So, you should clarify what you exactly expect from your study – it is definitely not the experience of ragging; in this case young students should have been enrolled. So, you should precisely define your scientific interest (e.g., to understand how ragging invades student life and culture). Of course, you can criticize the practice of ragging in the Discussion but you should not use the study participants for this purpose; they are much more ambivalent.

As ragging is a very sensitive issue in Sri Lanka, we had to be very cautious in our selection of participants. We also realized that we had not mentioned that perpetration of ragging occurs from the second year of university attendance, onwards and have now included this information in the introduction (Page 5, line 109-111). In order to better define/clarify our scientific interest, we have amended the aim (Page 6, line 141-143) of our study. 

This paper is a part of a larger study on ragging where we have found the prevalence rates of ragging among students at Jaffna University to be over 50%. We have another qualitative manuscript exploring the perceptions of lecturers and other key persons attached to the University highlighting that angle.

2. My second major concern refers to what you define as the 'overarching theme' of your analysis: "ragging as a means of communication". It was you, not the students, who stimulated a communication about ragging. Even more important, the students first reaction in the group discussion was to emphasize that ragging did not occur (line 391). It seems to be a secret or hidden practice, but definitely not a ‘means of communication’. Perhaps you mean something else but your expression is in any case misleading. By the way, I’m not sure whether you need an overarching theme at all. Isn’t it sufficient to define your subthemes as main themes and to present them as your results?

Thank you for your comment. Following a discussion between the authors, we have decided to change the overall theme to “Ragging as an expression of power” which maybe better suited for the study. (Fig. 2. Main theme and subthemes and page 11, line 277)

In this case, it may be a good idea to start the Results section with the issue of 'secrecy' and your overall observation that all groups denied ragging in the beginning of the discussion. This is really striking, given the formative experience of ragging. Obviously, older students try to forget or deny it. This could be the first important result and could later lead to an interesting discussion how difficult it is to make ragging 'public'.

We have made changes in the order of the results section to start with the “Veil of secrecy and silence” as it is an important and interesting aspect of ragging, as you underscored in your comments (Fig. 2. Main theme and subthemes and page 11, lines 288-309). 

3. Please check the logic of your themes and how you present them in the Results section. I see a couple of inconsistencies so that a re-arrangement may be required. 

Thank you for your comment. We have revised the order of the results in a manner that might be easier to follow. (Fig. 2. Main theme and subthemes)

(a) ‘Trivializing violence’ (lines 415 ff.) is not a subtheme of "Veil of secrecy and silence".

We have changed “Trivializing violence”, to belong to the subtheme “A society with deep divisions” as it is a culturally learned response. (Fig. 2. Main theme and subthemes and page 22, line 620-635).

(b) Although the examples under the subtheme "Rigid norms" are interesting, I’m not sure whether all of them belong to the main theme "A society with deep divisions". They deal, for example, with things like a general lack of protection of women (why is this a matter of ‘rigid norms’?). I think, other interesting examples, too, have nothing to do with ‘rigid norms’, such as equalizing (lines 503 ff.) or denying ethnic clashes (lines 536 ff.). Please check the main theme and the subthemes so that they are consistent. 

We have thought through your suggestion and revised the subthemes as well as removed some sections which we agreed could lead to confusion. Societal/Cultural norms are the foundations of how people comport themselves in groups. When there is an inflexibility between groups, there tends to be inequalities and higher rates of conflict. That said, the lack of protection for women occurred in small pockets and as it was not mentioned widely in the study so we decided to omit it. (Fig. 2. Main theme and subthemes)

(c) It is also not clear how the students’ feeling not to be heard by the society and the university (lines 542 ff.) or the feeling of insecurity (lines 571 ff.) is related to ragging. It is surely an important aspect, especially in the life of minorities but the authors should help readers to understand why these feelings and experiences may lead to ragging or may cause some students to accept ragging or to deny it. This should happen at least in the discussion.

Thank you for this observation. We realize that insecurity, might not be directly related to ragging and we have removed these sections. (Fig. 2. Main theme and subthemes)

(d) I would recommend to display “The students’ recommendations” (lines 585 ff.) as a main theme in Figure 2 and to present it as a main theme in the text so that readers can better follow your complete analysis and identification of themes.

We appreciate the reviewer’s suggestion and agree that it would be better to have “student recommendations” as a main theme. (Fig. 2. Main theme and subthemes and pages 24-25, lines 668-840)

4. The Discussion needs much more structure. You should start with a short and precise “summary of main findings”, no longer than 8 to 10 lines. I would suggest to stress the aspect of ambiguity or ambivalence towards ragging as the most important aspect (at least for me). The first sentence (lines 611 f.) Is superfluous; the last sentence (lines 617 ff. should be moved to the "Strength and Limitations" section in the Discussion. After the short summary you could go on with three major sections perhaps clearly marked with a subheading:

We have made changes to streamline the discussion as per your suggestion, and added subheadings to demarcate the three major sections for better flow. We combined the positive and negative ragging under the subheading of “Spectrum of ragging”. (page 25-31)

- Why ragging is positively perceived? (page 30, line 1125-1158)

- Why ragging is negatively perceived? (page 31, line 1160-1168)

- The wider context of ragging: Sri Lanka and its university culture (page 26, line 1002-1122)

5. I like your theoretical model of violence displayed in Figure 1, but I see two problems. I miss an “integration” as you claim with the Figure. Some arrows in the Figure and the pure statement “structural violence, intersectionality and social dominance theories are intricately linked” in the text (line 144) are not an integration. Moreover, you should use this model in a more creative way to explain ragging in societies like Sri Lanka. For example, as far as I understand, you say Sri Lanka is a violent country and unsatisfactory conditions contribute to student violence (line 657 f.). I agree, but one could ask, how? You say, ragging is violence and at the same time, ragging seems to be protest against inequalities and violence of the society. It is also remarkable that it is not the upper-class students (or only a few) who practice ragging but those at the other social end. Therefore, I would very much appreciate if you make a second attempt to use the model for an explanation.

We agree that there was much room for improvement in our model to demonstrate the integration of the three theoretical frameworks. These changes will be visible in Fig.1. Integrated theories on ragging in society.

6. In addition to reference 22, you may also refer to this just published study that shows the strong association of ragging and suicide: 

Thank you for the reference. We have included it in our manuscript (Reference 24, page 5-6, line 131-133).

7. Did you attach the COREQ statement for qualitative studies and did you fill in the page numbers, as requested?

We did not attach the COREQ guide statement for qualitative studies although we followed the guidelines. We have now attached it under supporting information. (S_2. File)

8. Line 399: One or two examples how seniors named juniors could be helpful to better understand this humiliating procedure.

We appreciate the suggestion to include an example of how the seniors humiliate the juniors and have included it (page 11-12, line 295-309 and page 13, line 333-343).

9. It would be generally helpful to give more examples (in form of an Appendix) how ragging is executed. It reminded me of practices and rituals of student leagues (“Burschenschaften) in Germany. Here, too, the younger ones have to be servants of the older students, are treated very badly, much alcohol is on the way and for some student leagues it is a special pleasure to hurt one another, using swords and, thus, to demonstrate masculinity. These practices are well known and also documented via media so that we have a very good idea of these rituals. Perhaps you have also some material to present examples in an Appendix that makes ragging easier imaginable for Non-Sri Lanka people.

Thank you for this recommendation to include examples of ragging to better understand this practice especially for non-Sri Lankans. We have added several newspaper articles and YouTube links as a supporting document. (S_3 File)

10. Line 729 ff. Of course, the practices you describe here are humiliating women. But please, consider that it can also be a hard expectation towards male students to behave in such an inhuman way. I do not want to exonerate males from accusations but some of them may also feel in a dilemma.

We share your belief that gendered expectations harm both women and men and we have rephrased this sentence (page 28, line 1077).

11. Line 785: I agree with your last plea but, following the students’ experiences, should it not read that it needs deep reforms in the university and the society to successfully combating ragging?

Thank you for your suggestion, which is also our belief and does strengthen our closing statement (page 33, line 1379-1381).

---

## [Editor Report · Decision Letter 3]

25 May 2022

PONE-D-21-15111R3“When a person can’t obey, he is subjected to violence”; a qualitative study on students perceptions on the phenomenon of ragging at a Sri Lankan universityPLOS ONE

Dear Dr. Wickramasinghe,

Thank you for submitting your manuscript to PLOS ONE. After careful consideration, we feel that it has merit but does not fully meet PLOS ONE’s publication criteria as it currently stands. Therefore, we invite you to submit a revised version of the manuscript that addresses the points raised during the review process.

From my point of view, the manuscript has clearly gained in structure, methodological clarity and explanatory power. Before the final acceptance of the paper and publication, I would like to suggest a number of changes and additions, which you hopefully appreciate:

1. It was a good idea to change the title, especially since the previous title was difficult to understand for readers who are unfamiliar with the background of Sri Lanka. However, I’m afraid, readers will also have problems to understand the meaning of the new title and its association with ragging, apart from the fact that “he” in the title is suboptimal and “(s)he” is not accurate … What do you think about including the topic of ‘power’ in the title, for example this way:

“Ragging as an expression of power in a deeply divided society — a qualitative study on students’ perceptions of ragging at a Sri Lankan university”

You may even find a better alternative. Please, consider, I’m a non-native speaker …

2. Although you tried to adapt the Abstract to the new version, it reflects the manuscript only to a limited degree. For example, main themes and sub themes are not well presented in the Abstract; also, the Discussion in the paper is not represented in the Abstract.

First of all, I would suggest to structure the Abstract with the help of the usual subheadings, i.e., Introduction, Methods and so on.  Please be aware, PlosOne allows 300 words (you have currently about 250 words).   

The last two sentences of the Abstract are very similar, so one of them could be deleted. Instead, there would be some more room for 1 or 2 sentences of a "Discussion ". Moreover, to be more precise, I would suggest to change one of the reported results in the Abstract this way:

INSTEAD:

The findings revealed how students used ragging as an expression of power to initiate order and as a way to express dissatisfaction towards social inequalities occurring within the larger society as well as facilitate bonds between university students.

BETTER:

The findings revealed how students used—or perceived—ragging very differently: as an expression of power to initiate order or as a way to express dissatisfaction towards social inequalities occurring within the larger society or to facilitate bonds between university students.

3. line 117: the word “ragging” is missing, isn’t it? (my “lines” always refer to the ‘clean’ version.)

4. For me, the lines 120 to 127 are crucial because they present and justify the Aim of the Study. However, I do not find the passage completely convincing and could imagine the following passage instead, based mainly on regrouping the sentences:

INSTEAD:

It is increasingly imperative to address this serious public health problem that profoundly affects all students, not only victims but also perpetrators and by-standers. Ragging has a potential deleterious impact upon society’s younger generations and their university years intended for building intellectual capacity. Educating youth in a safe space is essential, particularly for its subsequent contributions towards the country’s future. The aim of this study was to explore the perceptions of students concerning the phenomenon of ragging, and to understand how ragging affects student life and culture at the University of Jaffna.

BETTER:

Educating the youth in a safe space is essential, particularly for their subsequent contributions towards the country’s future. It is increasingly imperative to address this serious public health problem from the students’ perspective and to understand how it affects all students, not only victims but also perpetrators and by-standers. Since ragging has a potentially deleterious impact upon society’s younger generations and their university years intended for building intellectual capacity. The aim of this study was to explore how students perceive ragging and how ragging affects student life and culture at the University of Jaffna.

You may even find better alternatives …

5. From the history of the submission, I know that the chapter “Theoretical lenses” was desired by the reviewers for the Introduction. Although interesting, it may surprise the reader at this point, because after the formulation of the Aim of the Study, the Methods chapter is expected. That said, it would be much wiser to present the Theoretical lenses right there, namely in the Methods chapter, preferably starting at line 227 ff. What do you think?

While moving the Theoretical lenses into the Methods chapter, you should make it a bit clearer at this point that you used the Theoretical lenses to sensitize yourself for the data analysis. You have already mentioned this very well at the end of the Discussion; however, this argument should (also) come up already at the beginning of the Theoretical lenses chapter so that readers are informed about the significance of this chapter.

6. Due to the many (sub)headings in the Results chapter, it is not always clear which of them are the four main topics. I would therefore simply suggest (for lines 251, 278 etc.) to add: Subtheme 1: …, Subtheme 2: …  etc.

7. Line 283: I would suggest to start the first sentence with: For many students, ragging was part of …

8. Of course, I do not want to go into the details of your analysis. However, the presentation of the themes is still not quite logical in some places — despite the otherwise very successful revision. This is most noticeable in the case of sub-theme 2. For me, there are two very different topics that are shoved into each other here, namely ‘Ragging lies on a wide spectrum’ and ‘ragging is a cyclical event that thereby establishes hierarchies’.

While the lines 283 ff. belong to the ‘spectrum’ topic, lines 288 ff. belong to ‘hierarchy’. Lines 307 ff. deal again with ‘spectrum’ over a rather longer period. Lines 352 ff. then deal with ‘hierarchy’; lines 371 ff. clarify in particular the cyclic character, lines 386 ff. the ‘hierarchy’. Lines 406 ff. again the ‘spectrum’, in this case, the positive side. Lines 417 ff. deal again with the ‘cyclic hierarchies’. This is the way I read/understand your quotes.

I urgently ask you to check here again structure and argumentation thoroughly, as well as the naming of the main topics. Perhaps a fifth sub-theme will be appropriate to avoid the confusion in presenting sub-theme 2. In any case, your terms/themes should be distinctive, convincing and simply to follow for readers. 

We look forward to receiving your revised manuscript.

Kind regards,

Wolfgang Himmel

Academic Editor

PLOS ONE
---

## [Author Response · Author response to Decision Letter 3]

8 Jun 2022

Dear Wolfgang Himmel, 

Thank you for your very thorough reading of our manuscript. We appreciate you taking the time as the editor, and for all the insightful comments and recommendations you have made. We you are grateful for the opportunity to refine our manuscript to PLOS ONE standards. 

Changes made within the manuscript are highlighted below in blue along with a point-by-point response to your comments and concerns. All page numbers refer to the revised manuscript file with tracked changes. 

1. It was a good idea to change the title, especially since the previous title was difficult to understand for readers who are unfamiliar with the background of Sri Lanka. However, I’m afraid, readers will also have problems to understand the meaning of the new title and its association with ragging, apart from the fact that “he” in the title is suboptimal and “(s)he” is not accurate … What do you think about including the topic of ‘power’ in the title, for example this way:

“Ragging as an Expression of Power in a Deeply Divided Society; — a qualitative study on students’ perceptions of ragging at a Sri Lankan university”

Thank you for your suggestion. We agree that the title you suggested, better captures the understanding of our manuscript’s content and have revised the title accordingly. (Page 1 line 3-5).

2. Although you tried to adapt the Abstract to the new version, it reflects the manuscript only to a limited degree. For example, main themes and sub themes are not well presented in the Abstract; also, the Discussion in the paper is not represented in the Abstract.

First of all, I would suggest to structure the Abstract with the help of the usual subheadings, i.e., Introduction, Methods and so on. Please be aware, PlosOne allows 300 words (you have currently about 250 words). 

The last two sentences of the Abstract are very similar, so one of them could be deleted. Instead, there would be some more room for 1 or 2 sentences of a "Discussion ". Moreover, to be more precise, I would suggest to change one of the reported results in the Abstract this way:

INSTEAD:

The findings revealed how students used ragging as an expression of power to initiate order and as a way to express dissatisfaction towards social inequalities occurring within the larger society as well as facilitate bonds between university students.

BETTER:

The findings revealed how students used—or perceived—ragging very differently: as an expression of power to initiate order or as a way to express dissatisfaction towards social inequalities occurring within the larger society or to facilitate bonds between university students.

We appreciate your suggestions on refining the abstract for more clarity. 

Your observation of the repetition in the final lines of the abstract prompted us to delete one of the last sentences. (Pages 2, line 42)

We agree that it would be better to rephrase the findings in the abstract as recommended by the reviewer. (Page 2, lines 36-39)

While we appreciate the suggestion to include a structured abstract with subheadings, most qualitative work published in PLOS ONE had unstructured abstracts and we thought it was best to adhere to the general rule.

3. line 117: the word “ragging” is missing, isn’t it? (my “lines” always refer to the ‘clean’ version.)

We have rectified the omission and now included the word “ragging” (Page 5, line 127).

4. For me, the lines 120 to 127 are crucial because they present and justify the Aim of the Study. However, I do not find the passage completely convincing and could imagine the following passage instead, based mainly on regrouping the sentences:

INSTEAD:

It is increasingly imperative to address this serious public health problem that profoundly affects all students, not only victims but also perpetrators and by-standers. Ragging has a potential deleterious impact upon society’s younger generations and their university years intended for building intellectual capacity. Educating youth in a safe space is essential, particularly for its subsequent contributions towards the country’s future. The aim of this study was to explore the perceptions of students concerning the phenomenon of ragging, and to understand how ragging affects student life and culture at the University of Jaffna.

BETTER:

Educating the youth in a safe space is essential, particularly for their subsequent contributions towards the country’s future. It is increasingly imperative to address this serious public health problem from the students’ perspective and to understand how it affects all students, not only victims but also perpetrators and by-standers. Since ragging has a potentially deleterious impact upon society’s younger generations and their university years intended for building intellectual capacity. The aim of this study was to explore how students perceive ragging and how ragging affects student life and culture at the University of Jaffna.

You may even find better alternatives …

Thank you for your suggestion. We have revised the aim, taking your suggestion into consideration and rearranged the sentences to make the aim more succinct. (Page 6, lines 133-139). 

5. From the history of the submission, I know that the chapter “Theoretical lenses” was desired by the reviewers for the Introduction. Although interesting, it may surprise the reader at this point, because after the formulation of the Aim of the Study, the Methods chapter is expected. That said, it would be much wiser to present the Theoretical lenses right there, namely in the Methods chapter, preferably starting at line 227 ff. What do you think?

While moving the Theoretical lenses into the Methods chapter, you should make it a bit clearer at this point that you used the Theoretical lenses to sensitize yourself for the data analysis. You have already mentioned this very well at the end of the Discussion; however, this argument should (also) come up already at the beginning of the Theoretical lenses chapter so that readers are informed about the significance of this chapter.

Thank you for your comment. We have thought through your suggestion and made some changes to both the Theoretical lenses section (Page 6, lines 143-144) and the Methods section (Page 10, lines 242-243) which will enable the reader to comprehend this section better. 

Since it is common practice in qualitative studies to include the theoretical lenses at the end of the introduction, we have adhered to this format. Furthermore, theories most often evolve from the literature review and even when not made explicit, are ever-present throughout the work. For example, phenomenology is understood as foundational to qualitative approaches. There is an interplay between the theoretical lenses, the introduction to the literature, and the methods section. We believe, including the theoretical lenses at the end of the introduction, helps sensitize the reader early in the text thereby laying the groundwork in which to view the methods section. 

6. Due to the many (sub)headings in the Results chapter, it is not always clear which of them are the four main topics. I would therefore simply suggest (for lines 251, 278 etc.) to add: Subtheme 1: …, Subtheme 2: … etc.

We agree that including subheadings in the Results section would make the main topics/themes more clear (Page 11 line 267, Page 12 line 295, Page 15 line 380, Page 18line 464, Page 23 line 588). 

7. Line 283: I would suggest to start the first sentence with: For many students, ragging was part of …

We thank the reviewer for the comment and have made the necessary revision. (Page 12, line 296)

8. Of course, I do not want to go into the details of your analysis. However, the presentation of the themes is still not quite logical in some places — despite the otherwise very successful revision. This is most noticeable in the case of sub-theme 2. For me, there are two very different topics that are shoved into each other here, namely ‘Ragging lies on a wide spectrum’ and ‘ragging is a cyclical event that thereby establishes hierarchies’.

While the lines 283 ff. belong to the ‘spectrum’ topic, lines 288 ff. belong to ‘hierarchy’. Lines 307 ff. deal again with ‘spectrum’ over a rather longer period. Lines 352 ff. then deal with ‘hierarchy’; lines 371 ff. clarify in particular the cyclic character, lines 386 ff. the ‘hierarchy’. Lines 406 ff. again the ‘spectrum’, in this case, the positive side. Lines 417 ff. deal again with the ‘cyclic hierarchies’. This is the way I read/understand your quotes.

I urgently ask you to check here again structure and argumentation thoroughly, as well as the naming of the main topics. Perhaps a fifth sub-theme will be appropriate to avoid the confusion in presenting sub-theme 2. In any case, your terms/themes should be distinctive, convincing and simply to follow for readers. 

We appreciate your perspective here. We have revised the sub-theme titles by moving the section “Ragging lies on a spectrum” and making it a new sub-theme. We agree that it does impart more distinctive data than having it as part of the Cycle of Ragging. This will facilitate better understanding for the reader. (Fig. 2. Main theme and subthemes and page 12, lines 295-299).

We have also added a few sentences to clarify certain sections that may have been a little unclear. (Page 12, lines 298-453)

As recommended, a section found under “Ragging lies on a spectrum” has been moved to a more appropriate place in the text under “The powerful perpetrator” for easier comprehension. (Page 17, lines 432-439)

The section “Juniors dependent on seniors” is meant to illustrate dependency indicating subservience in the hierarchy. As the initial part of this section with its accompanying quote may had led to some confusion, we have omitted it from our revised manuscript to make it less ambiguous. (Page 17, lines 442-453)

---

## [Editor Report · Decision Letter 4]

24 Jun 2022

Ragging as an expression of power in a deeply divided society ; a qualitative study on students perceptions on the phenomenon of ragging at a Sri Lankan university

PONE-D-21-15111R4

Dear Dr. Wickramasinghe,

We’re pleased to inform you that your manuscript has been judged scientifically suitable for publication and will be formally accepted for publication once it meets all outstanding technical requirements.

Kind regards,

Wolfgang Himmel

Academic Editor

PLOS ONE

Additional Editor Comments (optional):

Let me say that I appreciated your efforts to constantly improve the manuscript. I hope you also agree that the manuscript is now much clearer in its arguments, easier to read and will evoke a better understanding of Sri Lanka's student culture and the country's structural problems. Thank you!   

Additional journal comments:

This decision was made after a re-evaluation following an appeal to a reject decision issued after a first round of revisions. More information about the appeal process can be found at https://journals.plos.org/plosone/s/editorial-and-peer-review-process#loc-appeals
---

## [Editor Report · Acceptance letter]

30 Jun 2022

PONE-D-21-15111R4 

Ragging as an expression of power in a deeply divided society; a qualitative study on students perceptions on the phenomenon of ragging at a Sri Lankan university 

Dear Dr. Wickramasinghe:

I'm pleased to inform you that your manuscript has been deemed suitable for publication in PLOS ONE. Congratulations! Your manuscript is now with our production department. 

Kind regards, 

on behalf of

Professor Wolfgang Himmel 

Academic Editor

PLOS ONE